# Surviving a Double-Edged Sword: Response of Horticultural Crops to Multiple Abiotic Stressors

**DOI:** 10.3390/ijms25105199

**Published:** 2024-05-10

**Authors:** Wenjing Yan, Rahat Sharif, Hamza Sohail, Yu Zhu, Xuehao Chen, Xuewen Xu

**Affiliations:** 1School of Horticulture and Landscape Architecture, Yangzhou University, Yangzhou 225009, China; wenjingyan0024@163.com (W.Y.); rahatsharif2016@nwafu.edu.cn (R.S.); hamzasohail@yzu.edu.cn (H.S.); z2118031272024@163.com (Y.Z.); xhchen@yzu.edu.cn (X.C.); 2Joint International Research Laboratory of Agriculture and Agri-Product Safety, The Ministry of Education of China, Yangzhou University, Yangzhou 225009, China

**Keywords:** reactive oxygen species, salinity, drought, heat, cold, waterlogging

## Abstract

Climate change-induced weather events, such as extreme temperatures, prolonged drought spells, or flooding, pose an enormous risk to crop productivity. Studies on the implications of multiple stresses may vary from those on a single stress. Usually, these stresses coincide, amplifying the extent of collateral damage and contributing to significant financial losses. The breadth of investigations focusing on the response of horticultural crops to a single abiotic stress is immense. However, the tolerance mechanisms of horticultural crops to multiple abiotic stresses remain poorly understood. In this review, we described the most prevalent types of abiotic stresses that occur simultaneously and discussed them in in-depth detail regarding the physiological and molecular responses of horticultural crops. In particular, we discussed the transcriptional, posttranscriptional, and metabolic responses of horticultural crops to multiple abiotic stresses. Strategies to breed multi-stress-resilient lines have been presented. Our manuscript presents an interesting amount of proposed knowledge that could be valuable in generating resilient genotypes for multiple stressors.

## 1. Introduction

Extreme climatic events lead to drastic changes in agroclimatic conditions and, in the process, negatively impact the yield of horticultural crops [1]. These climatic changes exert abiotic stresses with severe yield penalties and could inflict severe food shortages [2]. Abiotic stresses mainly comprise high temperature; low temperature; drought; waterlogging; salt, metal, nutrient, and air pollution; and photooxidative stresses. The horticultural crop industry is economically important and the lifeline of world food security [3]. Due to their intolerant nature, horticultural crops cannot endure abiotic stresses and are more affected than agronomic crops [4]. The quality of horticultural products is negatively impacted by abiotic stressors that occur during postharvest handling [5]. Abiotic stresses often do not exist alone in natural ecosystems. The growth and survival stability of horticultural crops is dramatically decreased with the increasing number and complexity of stresses that simultaneously affect horticultural crops [6]. It has also been reported that two or more combined abiotic stresses may produce additive effects [7]. The concurrent existence of two or more stresses can produce a synergistic harmful effect, which can be regarded as new stress, causing extreme irreversible damage [8,9,10].

Contrary to the previous statement, synchronized multiple stresses can also cause less damage than single stresses. For instance, the research on the *Eucalyptus globulus* response to numerous abiotic stresses illustrated that the combined drought and high temperature activated the protective responses, which was not observed when the stresses came separately [11]. For instance, *Vitis vinifera* (grapes) grow in regions characterized by a continental climate, such as North China, and face a combination of drought and cold stress, which affects their productivity [12]. Increased incidence of Fusarium wilt in *Solanum lycopersicum* (tomato) under salt stress was found to be caused by more sporulation of the fungi under saline conditions. *S. lycopersicum* exposed to combined salinity and heat stress performs better than plants subjected to these stresses separately. It has been reported that drought conditions in England and Wales have resulted in higher incidences of common scab caused by *Streptomyces scabies* in *Solanum tuberosum* (potato) [13]. Multiple stresses may produce complex synergistic, antagonistic, or additive effects (Figure 1). Following the occurrence of simultaneous abiotic stresses in horticultural plants, they activate an array of physiological and molecular responses, enabling them to endure (Figure 1). However, in the case of stress-prone cultivars, they subjugate to the wrath of stresses, resulting in significant yield losses. Economically, crop losses due to extremes in the environment have risen steadily over the past several decades, and climate models predict an increased incidence of floods, droughts, and extreme temperatures [14]. Integrated climate change and crop production models project declines in the yields of major crops such as corn, wheat, and rice with serious ramifications on global food production this century. Despite the progressive increase in the production of major crops through germplasm development and agronomic practices since the 1960s, susceptibility to climate variability has risen owing to higher sowing densities, which increase competition for water and nutrients [14]. Since these abiotic stresses often comes in combination, the rapid development and adoption of climate-resilient crop genotypes is imperative to ensure global food security.

In this review, we summarized the previous research reporting the response of horticultural crops to multiple abiotic stresses. We highlighted the effect of simultaneous abiotic stress on physiological and molecular machinery. Besides that, the strategies to generate resilient lines against multiple abiotic stresses has been discussed. We believe our study will lay a theoretical foundation for future in-depth explorations on the responses of horticultural crops to multiple abiotic stresses.

## 2. Common Types of Combined Multiple Abiotic Stresses

### 2.1. Heat and Drought Stress

Since the industrial revolution, the mean global surface temperature has risen by 1.1 °C and is expected to increase by 2–5 °C by the late 21st century [15]. Experts are forecasting an elevated frequency of temperature extremes above 38 °C by 2100 [16], posing a significant threat to agricultural production. Plants have optimal environmental conditions for growth and will encounter high-temperature stress once the air temperature exceeds the threshold. However, extreme climatic phenomena often do not occur alone in nature. For example, Qaseem et al. reported that high temperature is often accompanied by drought [17]. Extreme high temperature accelerates soil evaporation and increases transpiration by eliciting the opening of the plant’s stomatal pores, thereby exacerbating drought stress [18]. The lethality of collective heat and drought stress damaging the crop yield has been reported by Balfagón et al. and Sinha et al. [19,20], causing a massive economic loss [21]. In recent decades, tree mortality has been linked to prevailing temperature rise and drought [22]. High temperatures can slow tree growth and development by intensifying the drought time and frequency [23,24]. Several studies in horticultural crops have touched on this fundamental phenomenon of combined heat and drought stress effect on yield [25]. For example, in the case of soybeans, the effects of drought, high temperature, and their combination during flowering and pod initiation stages were examined. Higher chlorophyll contents, along with increased transpiration rate, were recorded in the tolerant cultivar compared to the sensitive one. The higher transpiration rate significantly reduced the leaf temperature, which eventually resulted in a response to combined drought and heat stress [25]. Similarly, the collective action of heat water deficit significantly altered the nitrogenase activity in the nodules and hampered the above-ground growth of soybean [26]. Despite the promising results, some pitfalls need to be filled by conducting more research on this critical area. A thorough transcriptomic and metabolomic study is required to understand the complex response of horticultural plants to combine heat and drought stress.

### 2.2. Heat–Drought–Light and Salt Stress

In addition, high-temperature and drought stress can frequently be combined with high irradiance and high salinity [27,28]. Drought and heat stress can enhance the saturation of the photosystem reaction center, increasing the sensitivity to high irradiance. For example, the photosynthetic pigment content of olive plants under heat–drought–UV-B (Ultraviolet-B) stress was higher than that under single-drought stress [29]. The effects of salt stress can be exacerbated by concurrent dry and heat stress and incredibly high evaporation rates in arid and semi-arid regions, causing soil moisture deficit and salt accumulation at the soil surface [30,31]. This will have a more adverse effect on the growth and yield of horticultural crops. In a study of *Brachypodium distachyon*, it has been found that the combination of drought and salt and heat–drought–salt stresses had varying degrees of adverse effects on plant performance. The significantly altered parameters included stem length, which was reduced by 59.1% and 61.8%, biomass, which was reduced by 61.9% and 63%, and grain number, which was reduced by 73.2% and 82% [32]. Two different resistant grape varieties, Touriga Nacional (TN) and Trincadeira (TR) were subjected to triple stress of heat–drought–light and restricted CO_2_ fixation, which substantially decreased the operating quantum efficiency of PSII (photosystem II) photochemistry in both varieties. In TN, which shows a higher capacity for heat dissipation and for withstanding high light intensities, total glutathione levels increased significantly, and the light compensation point had a higher value in response to heat–drought–light triple stress [33]. There are still gaps in this area which could be covered by conducting comprehensive studies on model horticultural crops.

### 2.3. Heat and Waterlogging Stress

Rainfall intensity due to high temperature, measured in days or even hours, increases, thus stimulating the possibility of extreme short-term rainfall [34]. Since the 1960s, the proportion of high-temperature-led weather extremes within the three days preceding torrential rain during summer in China, the Indian subcontinent (India, Pakistan, Sri Lanka, and Bangladesh), southeast Asia, and even Europe has increased significantly at a rate of 2.51%/10 years (national average) [35]. The simultaneous high temperature and heavy rainfall events mean horticultural crops face the double-edged sword of heat and waterlogging stress. In China’s middle-lower Yangtze River region, high temperature frequently occurs in summer and autumn, accompanied by frequent rainfall [36], adversely affecting cucumber production, a dominant horticultural crop grown in this region. Cucumber, originating from the tropical rainforest in the Himalayas, has a shallow root system and is not tolerant to waterlogging [37]. Meanwhile, cucumber is thermophilic but not heat-resistant, while thermal damage can be caused by the ambient temperature exceeding 35 °C [38]. The root is starved of oxygen for respiration under waterlogging conditions, and respiratory intensity skyrockets at high temperatures, thus simultaneously damaging the cucumber plant below and above ground. Extreme high temperatures and severe waterlogging events also frequently occur in summer over North China. Ginger (*Zingiber officinale*), an essential horticultural crop in North China, is intolerant to high temperature and vulnerable to waterlogging stress [39]. Thus far, attention is lacking in regard to combined heat and waterlogging stress in horticultural crops. More research work is required to address the underlying response mechanism and emphasize developing lines resilient to the combo of waterlogging and heat stress.

### 2.4. Heat and Salinity Stress

High temperature and salinity often pose a threat and thus far have been reported in several horticultural crops, including tomato, citrus, and *Casuarina glauca*. In general, *Casuarina glauca* is tolerant to salt or high-temperature stress alone but unable to cope when they happen simultaneously [40]. Concomitant high-temperature and salt stress conferred significant protection to tomato plants, as they alleviated the effect of single salt stress [41]. High transpiration evoked by high temperature neutralized the physiological response of Carrizo citrange to salt stress by hastening the leaf Cl^−^ intake. On the other hand, combined high- and high-salt stress had enhanced or attenuated hormonal responses to regulate the complex responses to single stress [42]. The collective response of the plant to multiple stresses at the same time continuously varies depending on which stress comes first.

Additionally, studies have reported how horticultural crops respond to combined stresses containing drought and salt [43,44], drought and cold [45], etc. Despite the importance, studies are lacking and need attention to address the complex mechanisms underlying these stresses. The application of heat and salinity on tomato or soybean crops for extensive genetic and genomic studies would lay a platform to explore the intricate signaling mechanisms governing these stimuli.

## 3. Different Strategies to Study the Stress Response

Although horticultural crops are frequently subjected to combined multiple abiotic stress events, research related to this field is scarce. The following sections present the standard methods for studying horticultural crops’ response to multiple stresses.

### 3.1. Evaluation of Phenotypic/Morphological Changes

Abiotic stress exerts the most intuitive effect on plants, causing changes in biological characteristics [46,47]. Hence, precise and effective phenotyping is critical for understanding the complicated responses of plants against abiotic stresses. High throughput phenotypic analysis is a classical method to analyze the crop phenotypes’ correlation with allelic variation and external stresses [48]. Fine and multi-level phenotypes were obtained through phenotypic analysis, which highlighted the responses of tomato to this class of biological stimulants, further paving the way for future research [49]. Several other phenotypic traits can be observed to see if the plant is under stress. For instance, root length, leaf mass per unit, leaf pigmentation, height at maturity, flowering time, reproduction phenology, and seed size and number are some of the key traits that can be examine for early stress response. Flowering time is a good example of a crucial trait that has been shown to be both under genetic control and plastic. Under climate change, the temperature cues triggering the chain of events leading to flowering might cease to be reliable if they occur at the wrong time with respect to the lifecycle and ecology of the species. Such changes in cue, signal, or response schemes might thereby elicit maladaptive responses [50]. Although an important area, as it is, lack of attention is preventing the exploration of this area to address current problems.

### 3.2. Transcriptomic Analysis

Transcriptome analysis is a sound strategy to obtain the critical components of different pathways governing growth and stress balance in plants [51,52]. Despite the discrepancy in different stress mechanisms, the ideas regarding investigation by using transcriptome techniques are generally similar. The molecular mechanisms of interaction between stresses can be explained by comparing the transcriptome data of plants under single and combined stresses [53,54]. The *Quercus ilex* L. was exposed to salt and ozone co-stress. The salinity had a considerable impact on the expression of several transcripts, while the effect of short-pulse ozone on expression was minimal. However, short-pulse ozone was capable of strongly inducing the up-regulation or down-regulation of salt-resistant genes. Multiple differentially expressed genes were associated with stress sensing and signaling, cell wall remodeling, reactive oxygen species (ROS), sensing and scavenging, photosynthesis, and glucose and lipid metabolism [55]. To reflect the differential gene expression patterns more comprehensively and intuitively in tissues, technologies such as single-cell transcriptome sequencing and spatial transcriptome sequencing have emerged based on traditional transcriptome sequencing techniques [56,57]. Sun et al. conducted single-cell sequencing on 30,000 individual cells of Chinese cabbage exposed to high-temperature stress. This study resulted in the creation of a single-cell transcriptional map that illustrates the leaf’s molecular responses to high temperature [58]. Heat shock factor (HSF) transcription factors promote metabolic enzyme and heat shock protein gene expression under heat stress. The WRKY8 transcription factor ortholog in cabbage leaves was activated by heat stress specifically in the vasculature but suppressed in the mesophyll. Additionally, 390 additional genes exhibited contrasting stress responses in various tissues [58]. In leaves exposed to mild drought stress, single-cell data rather suggested that the shut-down of energy-prone processes occurs in the *Arabidopsis* mesophyll, while the activation of defense genes is orchestrated by the epidermis [59]. Different studies demonstrated that the mesophyll tissue plays a prominent role in leaf response to salt, heat, and drought stress. In rice, salt causes the proliferation, differentiation, and maturation of the mesophyll to be postponed, resulting in a reduced number of mesophyll cells [60]. Initially captured by single-cell profiling, the effect on mesophyll development was biologically confirmed, illustrating the power of single cell RNA sequencing (scRNA-seq) not only to detect transcriptome changes, but also to suggest changes in organ composition. In drought-stressed *Arabidopsis* profiled after plant rewatering, single-nucleus omics data suggest that three WRKY family genes (*WRKY41*, *WRKY75*, and *WRKY53*) were specifically induced in vascular cells, while WRKY8 was induced in trichomes [61]. Although barely any of these tissue- or cell-type-specific transcriptional changes has been biologically confirmed by other means, because single-cell analysis of stress responses is still in its early days, they offer a new glimpse into the complexity of plant–environment interactions and open up a whole new field for further exploration, particularly in horticultural crops. Spatial transcriptome sequencing of *Portulaca oleracea* revealed complete integration of C4 and CAM systems, which have long been considered distinct and incompatible. They also predicted the integration of C4 and CAM systems under drought conditions, which was, for the first time, a clear description of C4 + CAM photosynthetic metabolism [62]. Despite the limited reporting of these methods in studies on horticultural crops’ responses to multiple abiotic stresses, it is plausible to assume that advances in sequencing technologies will encourage further research to address the cell-specific response.

### 3.3. Proteomic Analysis

Meanwhile, proteomic analysis is also an effective method to explore the behaviors of horticultural crops in the context of multiple abiotic stresses. Proteomics is a powerful tool for identifying differentially expressed proteins and quantitatively analyzing diverse biochemical pathways containing plant stress-associated responses [63]. Comparative proteomic analysis of plants exposed to specific stresses allows explorations of multiple defense-relevant mechanisms. Lin et al. adopted a proteomic approach to obtain protein expression profiles in broccoli [64]. They found that a higher Rubisco protein level when exposure to high-temperature and waterlogging stresses could improve carbon fixation efficiency, provide sufficient energy, and strengthen broccoli’s waterlogging stress tolerance at 40 °C. Cheng et al. conducted a comprehensive proteomic analysis of *Brachypodium distachyon* L. seedling leaves in the context of polyethylene glycol-simulated osmotic stress and cadmium (Cd^2+^) stress [65]. They revealed that osmotic and Cd^2+^ co-stress caused more significant impacts on seedling growth, leaf physiological traits, and ultrastructure than single stress.

### 3.4. Metabolomics

Metabolomics has become vital for comprehensively understanding cellular responses to abiotic stresses. It constitutes a crucial complementary tool currently available for the genomics-assisted selection of plant improvement [66]. Metabolomics aids in the identification of different compounds, including stress metabolic by-products, stress signaling molecules, or molecules implicated in plant acclimatization responses, offering valuable information for investigating the responses of horticultural crops to abiotic stresses [67]. Metabolic fingerprinting, metabolite profiling, and targeted profiling are the effective methods applied in plant metabolomic studies [68,69]. The investigation of phenolic metabolites in kale under low temperature and ultraviolet-a radiation (UV-A) exposure showed that UV-A irradiation markedly improved the growth parameters of the plants. However, low-temperature stress inhibited the plants’ growth; the plant growth parameters responding to low-temperature and UV-A co-stress were identical. Low temperature and UV-A treatment augmented ROS levels in kale, boosted metabolite biosynthesis, and elicited phenolic antioxidant metabolite accumulation. These results demonstrated that optimum low temperature and UV-A treatment led to phenolic metabolite increase in kale, thus improving its nutritional quality [70]. It is, therefore, suggested that metabolomics has an indicative role in cultivating superior horticultural crops that acclimate to multiple abiotic stresses.

### 3.5. Omics Approaches

With the rapid development of research platforms and bioinformatics tools, multi-omics integrated analysis has gradually become a widely used method to explore the reactions of crops to multiple abiotic stresses (Table 1). Through transcriptome and metabolite profiling, Demirel et al. found that the abundance of transcripts encoding proteins that participated in the light-harvesting complex of PSII was reduced in stress-tolerant cultivars. At the same time, the expression patterns of genes related to plant growth and development, hormone metabolism, and primary and secondary metabolisms were all higher than in stress-sensitive cultivars [71]. Therefore, it was demonstrated that the potato’s tolerance to single and multiple abiotic stresses was linked to maintaining CO_2_ assimilation and protecting PSII by attenuating light-harvesting capacity. Combining transcriptomics with metabolomics and integrating the results of gene and metabolite profiling, Liu et al. found that cold stress up-regulated genes, including *PAL*, *CHS*, *COMT*, *CCR*, and *COMT*, leading to elevated coniferyl alcohol and eriodyctiols contents in *Tetrastigma hemsleyanum* [72]. In contrast, this enhancement was weakened by 75% hydrogen-rich water intervention.

## 4. Response Mechanism of Horticultural Crops to Multiple Stresses

Plants, due to their sessile nature, often endure stress. However, sometimes various stresses come simultaneously, testing the plant’s immune response to the limits. Below, we discuss the detailed mechanisms by which horticultural plants respond to multiple stresses.

### 4.1. Physiological and Biochemical Response

Abiotic stresses affect many physiological aspects of horticultural crops, giving rise to extensive changes in cellular processes. These changes can be classified into non-adaptive and adaptive responses. Non-adaptive responses can only reflect adversity-caused damage to horticultural crops. For instance, drought, salinity, and chilling induce osmotic effects on plants, resulting in the induction of standard physiological processes, one of which is an accumulation of osmoprotectants [86]. The other stress-induced response shared by almost all abiotic stress conditions is the production of ROS. Heat and salt stress are known to commonly affect the transport and compartmentation of ions in plants. Drought and salinity stress evoke the generic response of creating a physiological water deficit in plants. Additionally, both stresses cause decreased CO_2_ diffusion into chloroplasts due to reduced stomatal opening, which leads to reduced carbon metabolism [86]. High-temperature stress impedes membrane fluidity and disrupts protein structure [87]. Cactus shows structural modifications in morphology, physiology, and metabolism to acclimatize to high temperatures and drought [88]. Not a great deal of studies have been done to understand the physiological response of plants to multiple stresses.

#### 4.1.1. Antioxidant System

Antioxidant enzymes are often considered stress markers. In response to multiple abiotic stresses, horticultural crops have evolved several defense mechanisms, such as an antioxidant system, to tackle ROS-elicited oxidative stress [89]. ROS performs dual roles in horticultural crops against abiotic stresses, either as toxic byproducts of stress metabolism [90,91] or as essential signaling molecules that assist horticultural crops in acclimatizing to the environment [92,93]. Horticultural crops accumulate ROS under multiple abiotic stresses such as water deficit and high salt in combination with high light intensity or others, which can impair photosynthesis, enhance photorespiration, and alter normal cellular homeostasis, resulting in increased generation of ROS [94]. Dual stress (high temperature plus high salt) reduces horticultural crops’ growth and photosynthetic activity and evokes ROS accumulation [95]. However, excessive ROS causes damage to DNA, proteins, and membrane systems [96]. To prevent ROS damage, horticultural crops have evolved complex and precise antioxidant systems in which L-Ascorbic acid (AsA) and glutathione (GSH) act as vital antioxidants to scavenge excessive ROS. Two grape cultivars, Touriga Nacional (TN) and Trincadeira (TR), were subjected to simultaneous drought, high-temperature (HT), and strong-light stresses. It was found that the H_2_O_2_ content in TN, a more heat-tolerant cultivar, did not change under drought stress, decreased upon single high-temperature or strong-light treatment, and increased significantly after combined high-temperature and strong-light treatment. On the other hand, H_2_O_2_ content in TR decreased only after drought treatment but not under HT or light stress. Meanwhile, the total AsA level was higher in TN than in TR, and total GSH content was significantly elevated in both TN and TR exposed to the combination of these three stresses [33]. These data, firstly, demonstrated a relatively more robust stress tolerance of TN than TR, and secondly, suggested the roles of AsA and GSH in scavenging excessive ROS. It was not difficult to find that horticultural crops also had different working patterns in their antioxidant system response to single abiotic and multiple abiotic stresses. Furthermore, Lin et al. found that AsA might mediate the responses of tomato to combined high-temperature and waterlogging stress by stabilizing RNA transcription, protein structure, and metabolism [97].

#### 4.1.2. Hormonal Response

Hormones are endogenous biological molecules and the main drivers of growth, development, and the immune system [53]. Abiotic stress can alter the contents and activities of endogenous hormones in horticultural crops, impacting physiological and biochemical properties. Abscisic acid (ABA), cytokinin (CTK), auxin (IAA), brassinosteroid (BR), ethylene (ETH), gibberellin (GA), jasmonic acid (JA), salicylic acid (SA), strigolactone (SL), and melatonin (MT) are currently the major phytohormones reported for their protective role against multiple stresses [98,99].

ABA is a stress hormone that increases the resistance of horticultural crops mainly by closing stomata, maintaining water balance, and rising root permeability. Studies have demonstrated that ABA could regulate high-temperature, high-salt, and drought stresses [100,101]. Upon drought stress, salinization increased due to reduced soil moisture, further inflicting double stress on the plant. In response, ABA could control the cytoprotective turgor pressure on the plant epidermis, mediate stomatal closure, and thus prevent osmotic stress-elicited water loss [102]. However, stomatal closure can also avoid the transpiration of leaves, which may be detrimental to plant adaptation to high-temperature stress. Citrus plants exposed to dual stress (high temperature plus drought) revealed that even though drought evoked ABA accumulation in tissues, high-temperature stress alone, or combined with drought stress, inhibited ABA accumulation. It was suggested that citrus leaves had different programs for regulating ABA homeostasis under diverse abiotic stresses [81]. Moreover, Jia et al. found that ABA REPRESSOR1 was considerably up-regulated, whereas that of ABA RESPONSIVE ELEMENTS-BINDING FACTOR 2 was down-regulated in poplar roots under high-temperature plus drought stress [103]. The kiwi plant was subjected to combined drought and waterlogging stress [104]. ABA increased significantly under drought conditions compared with the control and waterlogged plants. ABA-related gene responses were substantially greater in roots than in leaves. ABA-responsive genes, *DREB2* and *WRKY40*, showed the most significant upregulation in roots with flooding and the ABA biosynthesis gene, *NCED3*, with drought. Two ABA-catabolic genes, *CYP707A i* and *ii*, could differentiate the water stress responses, with upregulation in flooding and downregulation in drought [104]. However, the regulatory mechanism of ABA concerning concomitant high temperature and drought remains unclear.

CTK is a class of hormones mediating the growth and development of horticultural crops in the presence or absence of abiotic stress [105,106]. Horticultural crops can increase their resistance to adverse abiotic factors by altering the concentration of endogenous CTKs or through their exogenous application [107]. Accumulating studies have shown the involvement of CTK in the responses of horticultural crops to multiple abiotic stresses. Mushtaq et al. adopted RNAi to down-regulate tomato’s cytokinin receptor gene, *SlHK2*, to evaluate its performance under single and combined stresses (high temperature and drought) [108]. The study demonstrated that down-regulation of *SlHK2* disrupted the cytokinin signaling pathway to improve the tolerance of tomato to synchronized drought and high-temperature stress.

Many studies have also demonstrated the critical role of IAA in controlling the responses of horticultural crops to osmotic stress [109], such as salt [110] and drought stress [111]. Zhou et al. reported that tomato leaves accumulate IAA responding to combined cold and drought stress [77]. Additionally, the IAA level in poplar leaves showed a decreasing trend under drought stress, but the additional treatment of high-temperature stress increased the concentration of IAA [112]. IAA could fine-tune the response of plants in a trade-off manner between growth and survival. However, solid evidence is required to prove our verdict regarding IAA’s role under multiple stresses.

SA accumulated in higher quantities in response to high-temperature and drought stresses [81]. This indicates a potential additive role of SA in citrus responding to dual stress (high temperature plus drought) and indirectly suggests that SA could be utilized as a biostimulant to protect plants from multiple stresses.

It was also interesting in the study of Balfagon et al. that JA and its derivative JA-Ile were accumulated in the citrus under the combined low-temperature and wounding stress [42]. Alternatively, under high temperatures and wounding stress, citrus did not accumulate JA or JA-Ile but accumulated SA.

Melatonin (MT) has been largely ignored in multiple abiotic stress studies in horticultural crops. Melatonin is considered a non-enzymatic antioxidant and could be vital in providing resistance against simultaneous stresses in horticultural plants [12,113]. To the best of our knowledge, only one study in the literature describes the importance of melatonin in mitigating simultaneous abiotic stresses [114]. In that study, the researcher explained how applying MT to tomato plants minimizes the damaging effects of heat and drought stress. The tomato plants treated with MT (150 μm) displayed resistance by regulating the stomatal conductance under combined drought and heat stress. ABA, a primary stress-responsive hormone and perhaps the leading actor in regulating stomatal conductance [115], was observed with no significant changes. Despite the evidence, we believe MT could be utilized as a protectant in the horticultural crops industry, which is vulnerable to the looming threat of combined abiotic stresses.

It has been proven that phytohormones such as ETH, GA, BR, and SL mediate the reactions of horticultural crops under individual abiotic stresses. Still, there are few reports on their regulatory mechanisms against multiple abiotic stresses. Exogenous application of 24-epibrassinolide (EBR) can induce the synergistic effect between the antioxidant enzyme system and ATP synthase β subunits, thereby improving the scavenging efficiency of ROS, which allows tomato to better respond to the combined low temperature and low light [116]. EBR also markedly increases aspartate, threonine, serine, glycine, and phenylalanine levels but represses the generation of cysteine, methionine, arginine, and proline in tomato, which can attenuate the combined deleterious effects of dual stress on photosynthesis and nitrogen metabolism [117]. Combined cadmium (Cd) and ozone (O_3_) interacted in an antagonistic manner rather than an additive or synergistic manner. Cd treatment reduced the ethylene emission but did not affect the photosynthetic process, while O_3_ fumigation augmented the ethylene emission and impaired the photosynthetic capacity [118]. The altered ethylene accumulation may assist the trade-off between cadmium/O_3_ stress and the photosynthetic process, thus preventing the plant from damage. Another study illustrated that multi-hormone crosstalk might alleviate oxidative stress caused by UV-B and dark treatments by regulating the synthesis of flavonoids and alkaloids. The SA and GA levels in *Mahonia bealei* were reduced under UV-B exposure but raised under UV-B and dark treatments, which may play a regulatory role in synthesizing alkaloids and flavonoids in M. bealei leaves. The higher accumulation of alkaloids and flavonoids can help reduce cellular oxidative stress and assist plant defense armory against multiple abiotic stresses [119].

Hormones are generally involved in regulating flower and fruit development. During the flowering stage, multiple stresses occur, such as heat + waterlogging stress and heat + aphids, as in the case of cucumbers. However, no such study is available to disclose the mechanism of the changes that occur in hormonal metabolism when stresses occur at the flowering and fruit development stages. Research is required to address these critical issues and significantly enhance the yield by reducing the damage inflicted by multiple abiotic stresses on pollens and flower/fruit development.

#### 4.1.3. Compatible Solutes

Compatible solutes are a group of biological macromolecules widely spread in the plant system that potently regulates response to multiple abiotic stresses. The accumulation of compatible solutes (betaine, polyols, amino acids, etc.) can increase intracellular water activity to reach average cell volume and turgor pressure levels, thus protecting horticultural crops from abiotic stresses [120]. Horticultural crops that accumulate osmoregulatory substances (proline, trehalose, mannitol, betaine, etc.) under abiotic stresses have better resistance [121]. The combined high-temperature and high-salt stress can attenuate high salt-caused damage in tomato through accumulated osmoprotectants (glycine, betaine, and trehalose) [41]. Mobilization of various metabolites in *Pleione aurita* (Orchidaceae) pseudobulbs, including amino acids and their derivatives, non-structural carbohydrates, phenolic acids, and flavonoids accumulated in abundance following drought stress and rewatering. The mobilization of these metabolites suggests that purine and phenylpropyl metabolism pathways play a role in the drought response of pseudobulbs [122].

A prior study elucidated that betaine could maintain the osmotic potential of plant cells [123]. A study of oleander with higher salt and drought resistance revealed that the superior resistance of oleander to moderate drought and high salt correlated with a higher accumulation of soluble carbohydrates and betaine in its leaves [124]. With the development of bioengineering, exogenous applications provide convenience for exploring the functions of betaine in stress response. Through the exogenous application of betaine, researchers found that betaine was crucial in mediating the reactions of horticultural crops to stresses (high temperature, low temperature, drought, high salt, heavy metals) [125]. In another example, the exogenous application of betaine elevated the photosynthetic rate and increased heat shock protein (HSP) expression and HSP accumulation in tomato exposed to high temperatures [126,127]. Betaine could enhance photosynthesis and stomatal conductance and reduce the photorespiration of tomato exposed to high salt stress [126]. Polyols contain sugar alcohols, myo-inositol, mannitol, etc. [128,129]. Sugar alcohols are crucial substances engaged in intracellular osmoregulation that can slow water loss during the drying process and retard water uptake during the rehydration, thereby playing a role in osmoregulation under high salinity, drought, and waterlogging [130,131]. In addition, sugar alcohols can enhance resistance by stabilizing proteins and scavenging toxic ROS [132]. Myo-inositol and its derivatives also regulate stress resistance, which can modulate osmotic balance and scavenge ROS and act as signaling molecules to mediate multiple metabolic pathways [133,134].

Similarly, mannitol exerts an important osmoregulatory role in the responses of horticultural crops to abiotic stresses [135]. Through a comparative stress response analysis in tolerant and susceptible cultivars of potato, more sugar alcohols, myo-inositol, and mannitol were found to be accumulated when subjected to high-temperature and drought stress [71]. Sugar alcohols, myo-inositol, and mannitol were documented to be collected in *Mentha piperita* and *Catharanthus roseus* responding to individual or combined high-temperature and drought stress, with the highest accumulation under exposure to the combined stress [136].

Many studies have unraveled that proline accumulation and plant tolerance are positively relevant to diverse abiotic stresses. Excessive proline within plant cells is conducive to maintaining cellular homeostasis, water uptake, osmoregulation, and redox balance, vital for repairing cell structure and relieving oxidative damage [137]. Significantly, the metabolic responses of horticultural crops under multiple stresses often differ from those under single stresses. Proline increased time-dependently in poplar under high-temperature or drought stress alone, but it increased from 0 to 12 h and then decreased significantly after 24 h in response to dual stress [138]. It was most likely that at the initial stage of dual stress, the poplar established novel cellular homeostasis by its defense mechanisms. However, the homeostasis in the poplar was disrupted upon prolonging the stress duration, leading to the inhibition of proline.

### 4.2. Molecular Response to Multiple Abiotic Stresses

Molecular components such as transcription factors and genes are often considered biological macromolecules that provide translational and post-translational responses to combined abiotic stresses [27]. These myriad molecules are involved in sensing environmental signals, the biosynthesis of key enzymes and hormones, and regulating trade-offs between growth and stress response.

#### 4.2.1. Signal-Sensing Mechanisms

Sensing environmental signals is the first step for horticultural crops against abiotic stresses. Stress sensors can remodel signal-transduction pathways and promptly initiate proper responses, allowing plants to convert abiotic stress stimuli into cellular signals [139]. Distinct from chemical signals, abiotic stress produces physical signals, primarily affecting all cell parts simultaneously. In other words, abiotic stress signals can be sensed independently by different positions of cells and different macromolecules and structures within cells [140]. Because the encoded sensor proteins are often encoded by family genes, a single gene cannot directly visualize the stress response phenotype owing to dysfunction [119]. Due to these factors, it is difficult for researchers to identify the key stress sensors in horticultural crops, especially those simultaneously subjected to multiple abiotic stresses. Recent studies on the sensing mechanisms of abiotic stress signals, calcium ions (Ca^2+^) sensors, and histidine kinases (HKs) have been shown to sense stress signals in various horticultural crops.

Ca^2+^ is a ubiquitous secondary messenger within horticultural crops that responds to multiple stresses and developmental processes. Ca^2+^ sensor is often considered the first-line response to external stimuli, triggering downstream signals [141]. Currently, four classes of Ca^2+^ sensors, consisting of calcineurin B-like proteins (CBLs), calmodulins (CaMs), calmodulin-like proteins (CMLs), and calcium-dependent protein kinases (CDPKs or CPKs), are identified in higher plants [142]. Several Ca^2+^ sensors perform a role in abiotic stress. For instance, tomato calcineurin-like B proteins (SlCBL1/2) are linked to drought stress responses [143]. The *AcoCPK6* and *AcoCPK3* in pineapple are sensitive to drought and salt stress. Their overexpression in Arabidopsis has significantly reduced seed germination rates and decreased root length and fresh weight compared to wild-type under high-temperature stress conditions [144]. Another study found that high temperature and salinity, along with oxidative stress, might cause an increase in the cytoplasmic free Ca^2+^ concentration in plants. Osmotic sensor hyperosmolality-gated calcium-permeable channel (OSCA) and its family member OSCA1 could hinder hyperosmotic stress-induced Ca^2+^ [145]. Based on this, OSCA1 can be regarded as a hyperosmotic sensor. It is worth mentioning that OSCA family members are also widely distributed in horticultural crops, such as cucumber (9 *CsOSCA*) and Solanum habrochaites (11 *ShOSCA*) [146,147,148]. We speculated that OSCA1 could also be a sensor in horticultural crops responding to hyperosmotic stress such as high temperature and high salt. Attention should be paid to the OSCA family in future studies on the sensors of multiple abiotic stress in horticultural crops.

HK as sensors are critical components of two-component systems, and several genes related to HK have been substantiated to show vital regulatory roles in responses to abiotic stress. For instance, *BrHK6*, *BrHK7*, and *BrHK8* in Chinese cabbage can negatively regulate drought stress [149]. *SlHK2*, *SlHK3*, *SlHK4*, and *SlHK5* in tomato are commonly down-regulated upon salt stress treatment, while *SlHKL2* is repressed following drought stress intervention [150]. *IbHK1a*, *IbHKL4*, *IbHK3*, *IbHK5*, and *IbHK1b* in sweet potato have positive functions under high-temperature stress [151]. *CsHK2* and *CsHK4* are down-regulated in cucumber upon high-salt stress. *CsHK6* and *CsHK4* are down-regulated in response to high-temperature stress [152]. Although HK in horticultural crops responding to multiple abiotic stresses have not been reported, the detailed role of this signaling gene is worth investigating. Owing to the lack of research, the opportunity is on the plate to thoroughly investigate the role of histidine kinase genes in model horticultural plants.

#### 4.2.2. Signal Transduction Mechanism

Protein phosphorylation, a crucial post-translational modification mechanism, is considered one of the main signal transmission mechanisms. Protein phosphorylation alters the levels of downstream genes and modulates other biological processes, contributing to the transfer and amplification of external signals [153]. Protein kinases represent a cluster of enzymes catalyzing protein phosphorylation, among which mitogen-activated protein kinases (MAPKs) are one of the most extensively studied gene families, responding to multiple stresses [3]. The MAPK cascade minimally comprises three reversibly phosphorylated kinases, including MAPK, MAPK kinase (MAPKK/MEK), and MAPKK kinase (MAPKKK/MEKK) [154]. These kinases are key downstream signaling modules of RLKs and function as molecular switches in sensing upstream signals and responding to environmental stresses [155]. These kinases also modulate relevant gene expression and phytohormone accumulation/ degradation, thereby playing a critical regulatory role in horticultural crops under adverse environments [156]. Most SlMAPKKK genes in tomato were detected to be remarkably up-regulated in the context of abiotic stresses (heat, cold, drought, and salt). The relative mRNA levels of *SlMAPKKK51*, *SlMAPKKK53*, and *SlMAPKKK53* were raised over 100-fold following high-temperature or drought treatment, while the changes in 13 MAPKKK genes induced by salt treatment were over 10-folds. The data above suggested the participation of most *SlMAPKKK* genes in mediating multiple abiotic stress signaling pathways [157]. MAPK protein in cucumber also participates in signal transduction in abiotic stress response. By examining the expression profiles of *CsMAPKs* under three abiotic stresses (cold, heat, and drought), all *CsMAPKs* except *CsMPK3* and *CsMPK7* were respectively up- and down-regulated following cold treatment and heat stress. Meanwhile, all *CsMAPKs* were down-regulated in the first two days after drought treatment and markedly up-regulated after that [158]. In addition, MAPK members have a role in signal transduction in horticultural crops such as *Vitis vinifera* and *Vicia faba* L., responding to abiotic stresses [159,160,161].

In recent years, with global climate change and increasingly severe environmental pollution, horticultural crops are facing more and more abiotic stresses. Among them, the MAPK cascade has been shown to play an essential role in response to environmental stresses.

Firstly, understanding the role of MAPK in lignin biosynthesis is crucial to dissect their involvement in signal transduction under stressful conditions. Lignin is a natural polymer widely present in plant cell walls. Lignin provides cell wall strength and stability and increases plant resistance to abiotic stresses. Recent studies have shown a link between the MAPK pathway and lignin biosynthesis in *Arabidopsis thaliana* [162]. Some enzymes involved in lignin biosynthesis may be regulated by MAPK signaling pathways, thereby affecting lignin content and composition, ultimately resulting in changes in plant response to the external environment. Further study on the role of MAPK in lignin biosynthesis would be helpful in better understanding the response mechanism of horticultural crops to abiotic stress.

Secondly, we must study other genes and signaling molecules associated with MAPK. In addition to MAPK itself, many genes and signaling molecules are associated with it, such as MAPK-activated enzymes and downstream transcription factors. The regulation of these genes and signaling molecules will influence the role of MAPK in plant responses to abiotic stress. Studying the interaction of these genes and signaling molecules with MAPK can help us understand the role of MAPK in response to multiple abiotic stresses in horticultural crops.

#### 4.2.3. Mechanisms of Gene Expression Modulation

The gene expression regulation involves genetic, transcriptional, post-transcriptional, translational, and post-translational modifications.

*Trollius chinensis*, when subjected to drought stress and rehydration, displayed WRKY and AP2/ERF as the most differentially expressed families. Along with WRKY and AP2/ERF, the chlorophyll synthesis-related genes were also observed in the differentially expressed genes. The results lead us to assume that WRKY and AP2/ERF work in conjunction to regulate Trollius chinensis’ response to drought and rehydration [163]. Zhu et al. performed RNAi silencing of *SlNAC4* and found it conducive to salt and drought stress [164]. Recent studies have further proposed significant roles of post-transcriptional gene expression regulation in mediating the response of horticultural crops to abiotic stresses, which mainly involve pre-mRNA splicing and processing, regulation of mRNA stability and its degradation process, etc. For instance, pre-mRNA alternative splicing can modulate the responses of tomato against high-temperature stress [165]. The Cys2/His2 (C2H2)-type zinc finger protein MdZAT5 targets drought-responsive RNAs and microRNAs to control apple’s tolerance to drought [166].

Many transcription/protein targets of post-transcriptional and post-translational modification usually are modulators of cellular processes, encompassing signaling components and transcription factors (TFs). Since transcription factors can control key downstream responses by modulating target gene transcription, they have been significant targets for elevating plant stress resistance. When TFs bind with specific cis-responsive elements in the promoters of stress-inducible genes, the entire signal transduction cascade is activated, thereby improving the combined tolerance of plants against multiple stresses [167]. Transcription factors are involved in the regulation of abiotic stress-related genes in plants. As the response signals are transmitted, transcription factors are activated and bind to response elements in the DNA. Here, they can act as transcriptional activators or repressors, directly affecting the expression of target genes. TFs such as MYB, NAC, WRKY, and DREB are the most frequently occurring gene families in the mechanistic investigation of transcriptional regulation in horticultural crops against multiple abiotic stresses. *XsMYB44* in yellowhorn regulates stomatal closure to improve its tolerance responding to high-temperature plus drought stress [168]. In pepper, *CaWRKY6* confers resistance to high temperature and high humidity via transcriptionally activating *CaWRKY40* [169]. Combined heat and drought stress activates banana stress NACs, most of which are regulated by ABA [170]. DREB2A and DREB2C are the key TFs of tea plants in response to combined cold and drought stress [171] (Figure 2).

Transcription factors can also interact with multiple abiotic stress response pathways since different abiotic stresses simultaneously affect multiple metabolic pathways in plant cells. Transcription factors play an essential role in this process. They can act as nodes downstream of the signal and regulate the interaction between multiple abiotic stress response pathways. For example, ROS play a wide range of regulatory roles in plant responses to multiple abiotic stresses. ROS can be produced in multiple abiotic stress signal transduction pathways, including high-temperature, drought, salt, and alkali stresses. Numerous transcription factor families, such as bZIP, MYB, and AP2/ERF, can respond to the regulatory effect of ROS and participate in the regulation of multiple abiotic stress response pathways [110,172,173].

## 5. Breeding Strategies Using Molecular Approaches

Traditional plant breeding methods encounter limitations in satisfying the needs of the ever-growing global population. Recent progress in marker-assisted selection and genomics-assisted breeding has expanded the limits of conventional breeding methods. Genome editing has revolutionized plant breeding by providing innovative technologies with unparalleled precision, efficiency, and cost-effectiveness. Several genome editing techniques are used to generate stress-resilient lines. Due to its relative ease of use, clustered regularly interspaced short palindromic repeats (CRISPR)/CRISPR-associated protein 9 (Cas9) has become the genome-editing technology of choice. RNA-seq technology or forward genetic screening can identify the candidate gene responding to multiple stresses. Following identification, CRISPR technology can be used to knock out the candidate gene. This is a fast and efficient way to produce multi-stress resilient lines.

Plant species have discovered advantageous genes that have pivotal functions in augmenting plant traits. These genes have become prime targets in breeding programs to improve cultivars by employing conventional and contemporary breeding methods, such as marker-assisted selection, genetic engineering, and genome editing. Nevertheless, time consumption is a significant drawback in developing improved varieties. Traditional approaches take over a decade from the initial cross to the final release of an enhanced variety. Given these challenges, breeders must integrate various technologies to accelerate crop development. Speed breeding technology is a revolutionary approach to cultivating plants that keep up with breakthroughs in genomic technology. Speed breeding enables the production of around three to nine generations each year, resulting in the rapid advancement and introduction of new crop varieties. The combination of CRISPR and speed breeding could be a potential strategy to generate lines tolerant to multiple stresses (Figure 3). Compared to traditional, transgenic, and genome editing breeding, the combination of speed breeding and genome editing can save around 4–5 years.

## 6. Conclusions and Future Perspectives

Extreme climatic events often inflict multiple abiotic stresses at the same time. Plants coping with simultaneous abiotic stresses adopt several strategies. Different physiological and molecular responses include antioxidant, hormonal, transcriptional, and genetic responses that govern plant immunity against simultaneous abiotic stresses. In addition, because horticultural crops include many types of vegetables, fruits, flowers, trees, etc., there is broad scope for researchers in this field to decipher the complex mechanisms of multiple abiotic stress tolerance. Stress-resistant horticultural plants can be generated through transgenic approaches that could be of value to guarding world food security.

Many distinct combinations of abiotic stresses on horticultural crops have been triggered by global warming, climate change, and environmental pollution. Although the mechanisms by which horticulture crops respond to single abiotic stress are generally understood, the consequences of combined stress on horticultural crops are complex. Furthermore, investigations of horticultural crops responding to numerous abiotic stresses typically investigate response mechanisms by manipulating stress variables artificially. In nature, horticultural crops subjected to multiple abiotic stresses frequently encounter biotic stresses and other unknown or unexpected challenges, complicating the breeding of stress-resistant germplasm.

Thus, developing relevant solutions for the above problems would greatly benefit from explorations on how horticultural crops respond to multiple abiotic stresses. Based on current studies, this paper considered starting with the following points: (1) To better understand the impacts of multiple abiotic stresses on horticultural crops, we can conduct studies on combined stresses through transverse and longitudinal comparisons. The combined and corresponding single stresses can be compared to clarify the regulatory factors and mechanisms. Longitudinally, studies on combined stresses can be carried out first on horticultural crops with a short growth period and small size of the genome, such as tomato and cucumber, etc., which can be then extended to more complex horticultural crops, contributing to quickly finding the breakthrough point of research and effectively shortening the duration of research. (2) Expediting the development of efficient research platforms and research technologies, such as novel remote sensing technologies, aids in the introduction of low-cost, high-throughput phenotyping and can capture a mass of genetic diversity. Furthermore, real-time monitoring of crops responding to multiple stresses in nature is also conducive to modifying the research plan and direction, which can quickly and effectively perform studies on horticultural crops responding to multiple abiotic stresses. In conclusion, studies on how horticultural crops respond to various abiotic stresses remain increasingly important, and understanding how they escape such stresses is crucial for developing stress-resistant horticultural crops.

## Figures and Tables

**Figure 1 ijms-25-05199-f001:**
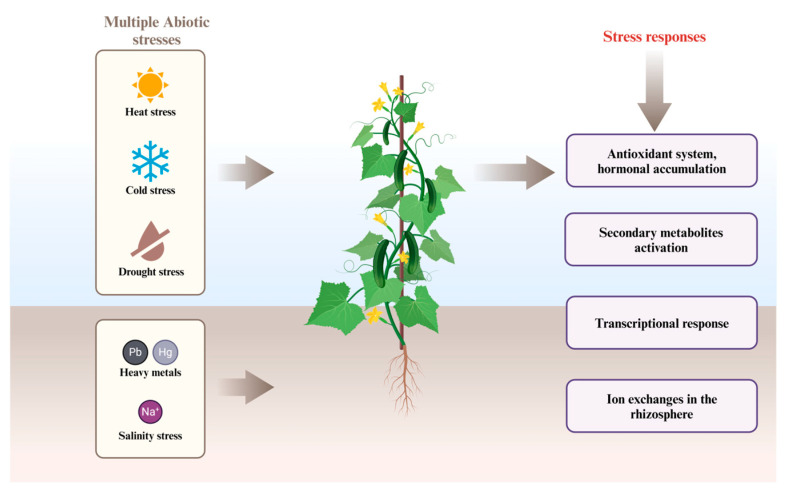
Illustration of different response mechanisms through which horticultural crops cope with simultaneous abiotic stresses.

**Figure 2 ijms-25-05199-f002:**
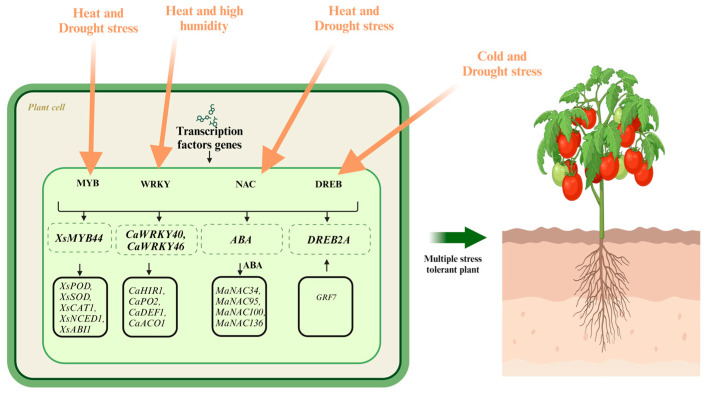
Transcriptomic response of horticultural crops to multiple abiotic stresses. Different transcription factors, including MYB, NAC, DREB, and WRKY, are depicted. In response to different combinations of multiple stresses, plants activate various TFs that bind to several stress-specific genes, further enhancing plant defenses.

**Figure 3 ijms-25-05199-f003:**
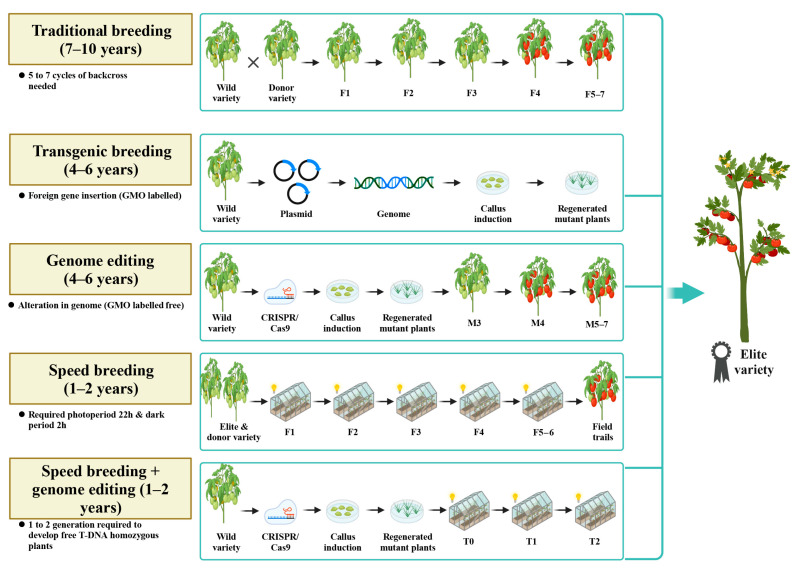
Schematic illustration of different breeding approaches. The conventional breeding technique, as well as the genome editing, is insufficient to meet the needs of growing world population amidst climate change. The combination of speed breeding and genome editing can save a number of years and can be useful to produce elite variety to increase the global food production.

**Table 1 ijms-25-05199-t001:** Enlisted studies relating to omics application under multiple abiotic stresses in horticultural crops.

Method	Species	Stress
Phenotypics	Tomato	Heat + Drought [49]
Transcriptomics	*Dianthus spiculifolius* Melon Pepper *Quercus ilex* Tea Plant Tomato	Cold + Drought [73] Heat + Humidity [74] Heat + Drought [75] Salt + Ozone [55] Cold + Drought [76] Cold + Drought [77]
Proteomics	Broccoli *Brachypodium distachyon* Mulberry *Portulaca oleracea* Potato	Heat + Waterlogging [64] Drought + Cd^2+^ [65] Salt + Drought [78] Heat + Humidity [79] Drought + Nitrogen [80]
Metabolomics	Citrus Kale Sugar beet	Heat + Drought [81] Cold + UV-A [70] Salt + Heat + Light [82]
Transcriptomics + Proteomics	Citrus	Drought + Heat + High irradiance [27]
Transcriptomics + Metabolomics	Date palm Grapevine Potato	Heat + Drought [83] Heat + Drought [84] Heat + Drought [71]
Proteomics + Metabolomics	*Brassica juncea*	Heat + Salt [85]

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
