# Peer review of "Surviving a Double-Edged Sword: Response of Horticultural Crops to Multiple Abiotic Stressors"

_ijms, 2024, doi:10.3390/ijms25105199_

Round 1

Reviewer 1 Report

Comments and Suggestions for Authors

Manuscript ijms-2967507 “Surviving a Double Edge Sword; Horticultural Crops Response to Multiple Abiotic Stressors”

 This paper presents a review where the whole-organism and molecular effects of multiple abiotic stresses on plants are discussed as well as the molecular mechanisms how plants survive multiple abiotic stresses. The topic of this manuscript is interesting and is relevant for IJMS. There are several issues that should be done before acceptance for publication to improve the manuscript and make it make this article more readable and easier to understand.

Recommended edits:

1) The manuscript would greatly benefit if to explain more clearly why suncronized multiple stresses may also induce less harm than single stress. I recommend to include more information and discussion on this, may be even in a separate subsection.

2) Title. It appears that semicolon is not appropriate here. I would change it to colon.

3) Section 3.1. Please add information what specific biological charactristics and phenotypic traits are usually analysed in phenotypic analysis for abiotiv stress effect evaluation.

4) Section 3.2. Transcriptomic analysis. Please add more examples on single cells transcriptomic analysis.

5) Latin names of plant species should be in italics. The manuscript contains many species that are not italicized.

- for example, line 38 Eucalyptus globulus, line 48 Streptomyces scabies, line 126 Casuarina glauca, line 130 Carrizo citrange, line 160 Quercus ilex, line 372 M. bealei., line 472 Solanum habrochaites, 541,545 Trollius chinensis

6) Appropriate reference should be added to:

Line 45,46.S. lycopersicum exposed to combined salinity and heat stress performs beĴer than plants subjected to these stresses separately.”

Line 292, 293. “ABA is a stress hormone…”

Line 541-542. “Trollius chinensis, when subjected to…”

6) Include author name to line 67-68. “The lethality of collective heat and drought stress damaging the crop yield has been reported by XXXXXX [18-19], ….”

7) It is necessary to put the abbreviations in order.

- line 87. UV-B is first time here. Full name also should be provided.

- line 99. PSII is firstly given here. Full name also should be provided.

- line 221. “Photosystem II” should be deleted here. See line 99.

- line 341. Melatonin should be MT here.

- line 369. Use only UV-B here without full name.

- lines 461, 468, 469 – calcium should be Ca2+ here. You have already provided full name above.

- line 477. Use HK instead of histidine kinases here. You have already provided full name above.

- line 581 Remove full name for ROS from here. It has been given above.

8) Line 226. Use lower register for CO2.

9) Check that gene names should be in italics thoroughout the manuscript.

e.g. line 228-229. PAL, CHS, COMT,  CCR,  and  COMT,

line 505 SlMAPKKK51,  SlMAPKKK53,  and  SlMAPKKK53

line 511, 512 CsMAPK

line 546 SINAC4

Comments on the Quality of English Language

10) English needs improvements.

- Line 29 “Abiotic stresses impede the yield and damage the quality of  horticultural produces during postharvest handling.” This sentence should be rewritten. It is not clear what did you mean here.

- line 37. Correct “Contrary to the previous statement, synchronized multiple stresses may also induce  

less harm than single stress.” To “Contrary to the previous statement, synchronized multiple stresses can also cause less damage than single stresses”.

- lines 92,93. Correct “In a study of Brachypodium distachyon found that the combination of drought   

and salt and the heat-drought-salt stress caused different degrees of detrimental impacts  on plant performances” to “In a study of Brachypodium distachyon, it has been found that the combination of drought and salt  and heat-drought-salt stresses had varying degrees of adverse effects on plant performance.”

- lines 94,95. Correct “Significantly changed parameters including culm length, which is  reduced by 59.1% and 61.8%, biomass reduced by 61.9% and 63%, and grain number reduced by 73.2% and 82%, respectively [31]” to “Significantly altered parameters included stem length, which was reduced by 59.1% and 61.8%, biomass, which was reduced by 61.9% and 63%, and grain number, which was reduced by 73.2% and 82% [31].

- line 96. Correct “In two different resistant grape varieties,…” to “Two different resistant grape varieties,….”

- line 141. Correct “As horticultural crops…” to “Although horticultural crops…”

- line 194. Correct “It revealed…” to “They revealed..”

- line 237. Correct “Below, we extensively discussed the detailed mechanism of horticultural crops to multiple stresses.” To “Below we discuss the detailed mechanism of how horticultural plants respond to multiple stresses.”

Author Response

1) The manuscript would greatly benefit if to explain more clearly why synchronized multiple stresses may also induce less harm than single stress. I recommend to include more information and discussion on this, may be even in a separate subsection.

Response: Thank you for highlighting this issue. We have addressed this matter in the manuscript. The detailed regarding why multiple stresses sometimes cause less damage than single stress is available in the introductions section line 38-50. Additionally, there is not enough literature available on the synergistic effects of combined stresses on plant in order to make a separate section. Therefore, we have mentioned the bits and bytes about the synergy between combined stresses and plant only in the introduction section.

2) Title. It appears that semicolon is not appropriate here. I would change it to colon.

Response: We have changed the title according to reviewer suggestions.

3) Section 3.1. Please add information what specific biological charactristics and phenotypic traits are usually analysed in phenotypic analysis for abiotic stress effect evaluation.

Response: We have added the details of phenotypic changes happens due to early abiotic stresses according to reviewer suggestion.

4) Section 3.2. Transcriptomic analysis. Please add more examples on single cells transcriptomic analysis.

Response: Thank you for highlighting this error. We have added additional studies regarding single cell transcriptomic analysis following reviewer suggestion. The additional studies related to single cell transcriptomic analysis were added in the section 3.2.

5) Latin names of plant species should be in italics. The manuscript contains many species that are not italicized.

- for example, line 38 Eucalyptus globulus, line 48 Streptomyces scabies, line 126 Casuarina glauca, line 130 Carrizo citrange, line 160 Quercus ilex, line 372 M. bealei., line 472 Solanum habrochaites, 541,545 Trollius chinensis

Response: Thank you for highlighting this issue. We have thoroughly revised the entire manuscript and italicized the names which were intended to be italicized.

6) Appropriate reference should be added to:

Line 45,46. “S. lycopersicum exposed to combined salinity and heat stress performs beĴer than plants subjected to these stresses separately.”

Line 292, 293. “ABA is a stress hormone…”

Line 541-542. “Trollius chinensis, when subjected to…”

6) Include author name to line 67-68. “The lethality of collective heat and drought stress damaging the crop yield has been reported by XXXXXX [18-19], ….”

Response: Thank you for highlighting this issue. We have added the names of author as per reviewer suggested.

7) It is necessary to put the abbreviations in order.

- line 87. UV-B is first time here. Full name also should be provided.

- line 99. PSII is firstly given here. Full name also should be provided.

- line 221. “Photosystem II” should be deleted here. See line 99.

- line 341. Melatonin should be MT here.

- line 369. Use only UV-B here without full name.

- lines 461, 468, 469 – calcium should be Ca2+ here. You have already provided full name above.

- line 477. Use HK instead of histidine kinases here. You have already provided full name above.

- line 581 Remove full name for ROS from here. It has been given above.

8) Line 226. Use lower register for CO2.

9) Check that gene names should be in italics throughout the manuscript.

e.g. line 228-229. PAL, CHS, COMT,  CCR,  and  COMT,

line 505 SlMAPKKK51,  SlMAPKKK53,  and  SlMAPKKK53

line 511, 512 CsMAPK

line 546 SINAC4

Response: We are grateful to the reviewer for highlighting these minor yet crucial mistakes. As they often get away from eyes, the effort of reviewer for thoroughly highlighting these are admirable.

10) English needs improvements.

- Line 29 “Abiotic stresses impede the yield and damage the quality of  horticultural produces during postharvest handling.” This sentence should be rewritten. It is not clear what did you mean here.

Response: Revision has been made as per reviewer suggestion.

- line 37. Correct “Contrary to the previous statement, synchronized multiple stresses may also induce less harm than single stress.” To “Contrary to the previous statement, synchronized multiple stresses can also cause less damage than single stresses”.

Response: Revision has been made as per reviewer suggestion.

- lines 92,93. Correct “In a study of Brachypodium distachyon found that the combination of drought  and salt and the heat-drought-salt stress caused different degrees of detrimental impacts  on plant performances” to “In a study of Brachypodium distachyon, it has been found that the combination of drought and salt  and heat-drought-salt stresses had varying degrees of adverse effects on plant performance.”

Response: Revision has been made as per reviewer suggestion.

- lines 94,95. Correct “Significantly changed parameters including culm length, which is  reduced by 59.1% and 61.8%, biomass reduced by 61.9% and 63%, and grain number reduced by 73.2% and 82%, respectively [31]” to “Significantly altered parameters included stem length, which was reduced by 59.1% and 61.8%, biomass, which was reduced by 61.9% and 63%, and grain number, which was reduced by 73.2% and 82% [31].

Response: Revision has been made as per reviewer suggestion.

- line 96. Correct “In two different resistant grape varieties,…” to “Two different resistant grape varieties,….”

Response: Revision has been made as per reviewer suggestion.

- line 141. Correct “As horticultural crops…” to “Although horticultural crops…”

Response: Revision has been made as per reviewer suggestion.

- line 194. Correct “It revealed…” to “They revealed..”

Response: Revision has been made as per reviewer suggestion.

- line 237. Correct “Below, we extensively discussed the detailed mechanism of horticultural crops to multiple stresses.” To “Below we discuss the detailed mechanism of how horticultural plants respond to multiple stresses.”

Response: Once again, we are grateful to the reviewer for highlighting the language related issues and guiding us to correct them. We appreciate the insightful peer review of this anonymous reviewer that allowed us to polish our manuscript and improves it readability.

Reviewer 2 Report

Comments and Suggestions for Authors

The review manuscript entitled “Surviving a Double Edge Sword; Horticultural Crops Response to Multiple Abiotic Stressors” describes the prevalent types of abiotic stresses that occur simultaneously and discusses them in in-depth detail regarding the physiological and molecular responses of horticulture crops. The manuscript presents a collection of proposed knowledge that could be used in generating resilient genotypes for multiple stressors. However, the role of multiple stresses in plants has been already over-reviewed, providing improving the manuscript it can go further publication processing. The manuscript needs a major revision. See the following comments:

1- The authors should review the existing literature and highlight specific areas or aspects of the topic that have not been adequately addressed or synthesized. This could involve focusing on emerging stressors, novel combinations of stresses, or underexplored physiological/molecular mechanisms.

2- Rather than just summarizing previous work, the authors could attempt to develop a new theoretical or analytical framework for understanding plant responses to multiple abiotic stresses. This could involve integrating different disciplinary perspectives or proposing innovative models.

3- The authors could strengthen the manuscript by clearly articulating the real-world implications of their review, such as informing crop breeding strategies, improving agricultural practices, or guiding stress management approaches.

4- If the authors can offer a particularly thorough, insightful, and well-structured synthesis of the current state of knowledge, this could justify the publication, even if the topic has been reviewed before.

5- To enhance the manuscript's impact, the authors should incorporate higher-quality, more detailed figures that illustrate the underlying molecular and cellular mechanisms of plant responses to multiple abiotic stressors. The current figures appear to lack the necessary visual clarity and depth of mechanistic insights to distinguish this review from the extensive existing literature on this subject. Incorporating figures that clearly depict key signaling pathways, metabolic changes, structural adaptations, or gene expression patterns at the subcellular level would bolster the manuscript and help justify its publication. Such informative visuals would provide a more compelling, in-depth synthesis of the current understanding, going beyond what has been covered previously. Integrating these types of detailed, insightful figures would enable the review to make a more valuable contribution to the field. I recommend using commercially available software and platforms like biorender.com.

6- The writing in the manuscript could benefit from a thorough language edit to improve the overall clarity, fluency, and precision of expression. Some sentences appear to have grammatical errors or phrasing that is not entirely idiomatic or natural-sounding in English. A careful review and editing of the language by a native English speaker or professional editor would help enhance the readability and scholarly tone of the work. Improving the language quality would ensure the key concepts and insights are communicated effectively to the intended audience.

7- Moreover, some references in the literature could be cited in this review paper:

Priya, P., Patil, M., Pandey, P., Singh, A., Babu, V. S., & SenthilKumar, M. (2023). Stress combinations and their interactions in plants database: a onestop resource on combined stress responses in plants. The Plant Journal116(4), 1097-1117.

Moradi, P., Vafaee, Y., Mozafari, A. A., & Tahir, N. A. R. (2022). Silicon nanoparticles and methyl jasmonate improve physiological response and increase expression of stress-related genes in strawberry cv. Paros under salinity stress. Silicon14(16), 10559-10569.

Comments on the Quality of English Language

The writing in the manuscript could benefit from a thorough language edit to improve the overall clarity, fluency, and precision of expression. Some sentences appear to have grammatical errors or phrasing that is not entirely idiomatic or natural-sounding in English. A careful review and editing of the language by a native English speaker or professional editor would help enhance the readability and scholarly tone of the work. Improving the language quality would ensure the key concepts and insights are communicated effectively to the intended audience.

Author Response

The review manuscript entitled “Surviving a Double Edge Sword; Horticultural Crops Response to Multiple Abiotic Stressors” describes the prevalent types of abiotic stresses that occur simultaneously and discusses them in in-depth detail regarding the physiological and molecular responses of horticulture crops. The manuscript presents a collection of proposed knowledge that could be used in generating resilient genotypes for multiple stressors. However, the role of multiple stresses in plants has been already over-reviewed, providing improving the manuscript it can go further publication processing. The manuscript needs a major revision. See the following comments:

Response: Thank you for the critical reviewing of our manuscript. Indeed, the role of multiple abiotic stresses has been presented previously, however, our manuscript is entirely centered upon the horticultural crops. Other than that, the cases of rice and Arabidopsis that we have provided highlight the gaps in the literature about horticulture crops. In addition, we have made extensive revisions to the entire work to enhance its readability significantly.

1- The authors should review the existing literature and highlight specific areas or aspects of the topic that have not been adequately addressed or synthesized. This could involve focusing on emerging stressors, novel combinations of stresses, or underexplored physiological/molecular mechanisms.

Response: Thank you for the comment. We have made revision to the section 3.1 “evaluation of phenotypic changes”. The more common one “heat and drought stress” section 2.1 has been highlighted several articles before but not specific to horticultural crops. The “Heat-drought-light and salt stress” section 2.2 has been rarely addressed previously. We put emphasis on it highlighted this multiple simultaneous stress effect on horticultural crops. The “heat and waterlogging” section 2.3 has been missing in the previously published review articles. Also, our lab is currently performing extensive research work on combining heat and waterlogging stress in cucumber. We believe less attention has been given to combine waterlogging and heat stress.   

2- Rather than just summarizing previous work, the authors could attempt to develop a new theoretical or analytical framework for understanding plant responses to multiple abiotic stresses. This could involve integrating different disciplinary perspectives or proposing innovative models.

Response: Thank you for the comment. This is a very important point highlighted by the reviewer. Our lab is currently working on the combined heat and waterlogging stress in cucumber. The work is rather complicated and in the meantime interesting as well. We already have identified some key transcription factors responding to this combo of stresses. In addition, we have specifically developed a model to simulate both waterlogging and heat stress, while keeping other factors such as drought, light, and moisture constant. The experiment is currently under progress and, upon completion, will be evaluated for potential publication to the International Journal of Molecular Sciences (IJMS).

3- The authors could strengthen the manuscript by clearly articulating the real-world implications of their review, such as informing crop breeding strategies, improving agricultural practices, or guiding stress management approaches.

Response: Thank you for the comment. We appreciate the reviewer that allowed us to improved our article by adding a new section 5. (Breeding strategies using molecular approaches). In this section we presented the potential of CRISPR genome editing technology and speed breeding. We believe the combination of CRISPR and speed breeding could facilitate the breeding programs of generating multi-stress resilient horticultural crops.

4- If the authors can offer a particularly thorough, insightful, and well-structured synthesis of the current state of knowledge, this could justify the publication, even if the topic has been reviewed before.

Response: Thanks to reviewer comment for encouraging us to polish our manuscript. We have made extensive changes and presented several new sections that we believe improved the quality of our manuscript.

5- To enhance the manuscript's impact, the authors should incorporate higher-quality, more detailed figures that illustrate the underlying molecular and cellular mechanisms of plant responses to multiple abiotic stressors. The current figures appear to lack the necessary visual clarity and depth of mechanistic insights to distinguish this review from the extensive existing literature on this subject. Incorporating figures that clearly depict key signaling pathways, metabolic changes, structural adaptations, or gene expression patterns at the subcellular level would bolster the manuscript and help justify its publication. Such informative visuals would provide a more compelling, in-depth synthesis of the current understanding, going beyond what has been covered previously. Integrating these types of detailed, insightful figures would enable the review to make a more valuable contribution to the field. I recommend using commercially available software and platforms like biorender.com.

Response: Thank you for the comment. We have drawn the figure 3 by purchasing and using the Biorender.com. We also enhanced the resolution of other figures.

6- The writing in the manuscript could benefit from a thorough language edit to improve the overall clarity, fluency, and precision of expression. Some sentences appear to have grammatical errors or phrasing that is not entirely idiomatic or natural-sounding in English. A careful review and editing of the language by a native English speaker or professional editor would help enhance the readability and scholarly tone of the work. Improving the language quality would ensure the key concepts and insights are communicated effectively to the intended audience.

Response: Thank you for the comment. I myself (Xuewen Xu) and Professor Xuehao Chen personally revised the manuscript a couple of times to minimize the number of mistakes in the writing. All the authors have extensively revised the manuscript several times. We are specially thankful to Dr Waqar Shafqat (Mississippi State University) for revising the language of manuscript. The certificate of English editing has been attached below.

7- Moreover, some references in the literature could be cited in this review paper:

Priya, P., Patil, M., Pandey, P., Singh, A., Babu, V. S., & Senthil‐Kumar, M. (2023). Stress combinations and their interactions in plants database: a one‐stop resource on combined stress responses in plants. The Plant Journal116(4), 1097-1117.

Moradi, P., Vafaee, Y., Mozafari, A. A., & Tahir, N. A. R. (2022). Silicon nanoparticles and methyl jasmonate improve physiological response and increase expression of stress-related genes in strawberry cv. Paros under salinity stress. Silicon14(16), 10559-10569.

Response: Thank you for the comment. We have cited the mentioned articles in the manuscript.

Reviewer 3 Report

Comments and Suggestions for Authors

Dear Authors,

I had opportunity to assess the review paper entitled :” Surviving a Double Edge Sword; Horticultural Crops Response to Multiple Abiotic Stressors” which is considered for publication in IJMS. Manuscript is generally interesting and rather good written but it should be focused mainly on molecular aspect of multistressors becouse the IJMS is focused on molecular aspects of organism as journal. The manuscript need also some improvements and rewriting. List of comments

The introduction Section

Need to present the directly and precisely the aim of review paper currently this part is fuzzy. The Figure 1 should be better described in this part of text now the figure is cited without logical description in text. Figure 1 please eliminate the drawings of crops this makes the figure not pretty and logical. The figures in review papers need to be eye catching.

Main part of review.

The review is not well balanced some fragments sections are extremely short like sections 2.1-2.4 and 3.1 and many more where others is long. This makes the reviews shredded to parts not exactly connected with each other in logical manner. In context and quality of IJMS also Aims of journal the section 4.2 must be enlarged. Is a lot of papers which present role of stressors especially with use of molecular approaches and transcriptome. Figure 2 is extremely low quality and difficult to read. The Figure captions should be self-descriptive in review paper. Authors must rethink and rewrite sections of review to make this more balanced I suggest to add new fragments to enlarge some paragraphs.

Sincerely,

Comments on the Quality of English Language

The text should be checked by native speaker it has spelling errors and need the grammar check

Author Response

I had opportunity to assess the review paper entitled :” Surviving a Double Edge Sword; Horticultural Crops Response to Multiple Abiotic Stressors” which is considered for publication in IJMS. Manuscript is generally interesting and rather good written but it should be focused mainly on molecular aspect of multistressors becouse the IJMS is focused on molecular aspects of organism as journal. The manuscript need also some improvements and rewriting. List of comments

Response: Thank you for the comment. I myself (Xuewen Xu) and Professor Xuehao Chen personally revised the manuscript a couple of times to minimize the number of mistakes in the writing. All the authors have extensively revised the manuscript several times. We are specially thankful to Dr Waqar Shafqat (Mississippi State University) for revising the language of manuscript.

The introduction Section

Need to present the directly and precisely the aim of review paper currently this part is fuzzy. The Figure 1 should be better described in this part of text now the figure is cited without logical description in text. Figure 1 please eliminate the drawings of crops this makes the figure not pretty and logical. The figures in review papers need to be eye catching.

Response: Thank you for the comment. We have rewritten the aims of our reviewer article following reviewer suggestion. Also, we have properly addressed the figure 1 in the main text. According to the reviewer’s suggestion, we have redrawn the figure 1 and believe that the revised version of figure is eye catching and informative.   

Main part of review.

The review is not well balanced some fragments sections are extremely short like sections 2.1-2.4 and 3.1 and many more where others is long. This makes the reviews shredded to parts not exactly connected with each other in logical manner. In context and quality of IJMS also Aims of journal the section 4.2 must be enlarged. Is a lot of papers which present role of stressors especially with use of molecular approaches and transcriptome. Figure 2 is extremely low quality and difficult to read. The Figure captions should be self-descriptive in review paper. Authors must rethink and rewrite sections of review to make this more balanced I suggest to add new fragments to enlarge some paragraphs.

Response: Thank you for the comment. We have extensively revised the entire manuscript. The language of the manuscript has been revised by Dr Waqar Shafqat (Mississippi State University). The section 4.2 has been fragmented according to reviewer suggestion. Yes we agree with the reviewer, the figure 2 quality was extremely poor and was hard to understand. The figure 2 has been redrawn and inserted in the revised manuscript with high resolution 1200 DPI. In addition, we have included a new section 5.0 which is presenting different strategies to maximize the yield in this rapidly changing climate.

Reviewer 4 Report

Comments and Suggestions for Authors

The authors summarized the relationship between abiotic stress and horticultural crops here with response mechanism. A few issues, in my opinion, that this article needs to resolve before publishing it:

1. Abstract is written with overall information. It does not represent the all the topics included in this ms.

2. Author needs to study the economic/market losses due to these stressess. Without mentioning economic part this study is incomplete

3. Line 28 is not fully correct. Please revise your thoughts regarding agronomic crop loses.

4. Figure 1 representing too shallow information. Make it with more thoughts and insights. The picture quality and background color is another concern need to work.

5. in paragraph 2.3: we need an idea of global perspective. Please rewrite

6. I highly recommend to draw a figure on RESPONSE MECHANISM to address all in an easier form to understand.

I am looking forward to see the revised ms.

Author Response

The authors summarized the relationship between abiotic stress and horticultural crops here with response mechanism. A few issues, in my opinion, that this article needs to resolve before publishing it:

  1. Abstract is written with overall information. It does not represent the all the topics included in this ms.

Response: Thank you for the comment. We have revised the abstract and added the details which were suggested by the reviewer.

  1. Author needs to study the economic/market losses due to these stresses. Without mentioning economic part this study is incomplete.

Response: Thank you for the comment. We have added information related to economics in the introduction section. However, we couldn’t find an exact figure of how much losses do multiple abiotic stresses inflict on farmer (USD$).  

  1. Line 28 is not fully correct. Please revise your thoughts regarding agronomic crop loses.

Response: Thank you for the comment. We have included information regarding the economic losses due to abiotic stresses in the introduction section. Yet again we cannot get an exact figure (USD$) in losses.

  1. Figure 1 representing too shallow information. Make it with more thoughts and insights. The picture quality and background color is another concern need to work.

Response: We have redrawn the figure 1 and properly addressed the it in the main text according to reviewer suggestions.

  1. in paragraph 2.3: we need an idea of global perspective. Please rewrite

Response: Thank you for highlighting this error. We have rewrite the statement and gave it a global perspective.

  1. I highly recommend to draw a figure on RESPONSE MECHANISM to address all in an easier form to understand.

Response: Thank you for the comment. We have redrawn all the figures since the previous version of them was not clear. Additionally, we have included figure 3 representing the combination of speed breeding and genome editing to tackle the multiple abiotic stresses, thus to ensure food security.

I am looking forward to see the revised ms.

Response: Thank you for the comment. We have extensively revised the entire manuscript. The language of the manuscript has been revised by Dr Waqar Shafqat (Mississippi State University).

Round 2

Reviewer 3 Report

Comments and Suggestions for Authors

Dear Authors/Editors,

All improvements was added. Irecpmend publication.

Sincerely,

Reviewer 4 Report

Comments and Suggestions for Authors

Loks better. Thank you